# Longitudinal trends in physical activity and sedentary behaviour among school children in Norway: The health oriented pedagogical project (HOPP)

Per Morten Fredriksen[1], Iana Kharlova[2], Maren Valand Hæhre[2], Asgeir Mamen[2]*

**1** University of Inland Norway, Elverum, Norway, **2** Kristiania University College, School of Health Sciences, Oslo, Norway

* asgeir.mamen@kristiania.no

## Abstract

### Introduction

Many children do not achieve recommended levels of physical activity (PA), although PA is important for cardiovascular health, mental well-being, and reducing chronic disease risk. The aim of this study was to describe longitudinal changes in moderate-to-vigorous physical activity (MVPA) and sedentary behavior (SED) during primary school and to examine whether a school-based active learning intervention was associated with attenuated age-related declines in PA. Baseline cross-sectional age gradients are presented as descriptive context and are not interpreted as a causal counterfactual for longitudinal change.

### Methods

The study sample consisted of n = 2123 children (50% females) from the Health Oriented Pedagogical Project (HOPP) in Norway with valid accelerometer data in 2015, aged 6–12 years (mean 9.46 ± 1.75). MVPA and SED were objectively assessed using hip-worn accelerometers based on seven-day averages. Data from 2015–2019 were analyzed using linear and generalized linear mixed-effects models to account for repeated measures and clustering.

### Results

Baseline cross-sectional analyses suggested a decline of 3.5 min·day$^{-1}$ of MVPA per year of age among children aged 6–12 years. Longitudinally, across five years (2015–2019), MVPA declined on average by 2.2 min·day$^{-1}$·year$^{-1}$ (p < 0.001) and SED increased by 6.7 min·day$^{-1}$·year$^{-1}$ (p < 0.001).

**Data availability statement:** The dataset contains de-identified individual-level data from children and cannot be made publicly available in an open repository due to ethical and legal restrictions related to participant confidentiality and the informed consent framework. A de-identified minimal dataset necessary to replicate the analyses can be made available through a controlled access procedure via the HOPP Data Access Committee, University of Inland Norway. Requests should be submitted to research director at Marte Tøndel (marte.tondel@inn.no) with a brief description of the proposed use. Access is subject to committee review and approval and completion of a data access/data transfer agreement in accordance with applicable GDPR and Norwegian regulations.

**Funding:** We are grateful to the following institutions that have contributed with financial support: Horten Municipality, Kristiania University College, the Norwegian Order of Odd Fellow Research Fund, the Oslofjord Regional Research Fund, and the Norwegian Fund for Post-Graduate Training in Physiotherapy. The funders had no role in study design, data collection and analysis, decision to publish, or preparation of the manuscript.

**Competing interests:** The authors have declared that no competing interests exist.

**Abbreviations:** CI, confidence interval; HOPP, The Health Oriented Pedagogical Project; MVPA, moderate-to-vigorous physical activity; PA, physical activity; SE, standard error; SED, sedentary behavior.

## Conclusion

Physical activity levels declined, **and sedentary** behavior increased with age, beginning in early primary school. Over follow-up, the observed longitudinal decline in MVPA and increase in SED were less steep than the baseline cross-sectional age gradients presented for context, but these comparisons should be interpreted descriptively rather than causally.

## Introduction

Physical inactivity is a major determinant of several non-communicable diseases (NCDs) and is commonly cited as one of the leading global causes of premature mortality [1,2]. Preventing NCD risk factors across the population therefore requires sustained attention to physical activity (PA), and the most effective window for establishing active habits is early in life, when children's movement is closely integrated with play, exploration, and development [2]. At the same time, there is increasing concern that many children do not achieve recommended activity levels, as the World Health Organization advises at least 60 minutes of moderate-to-vigorous physical activity (MVPA) daily for 5–17-year-olds [2].

During the preschool years, PA may increase as motor competence develops, and studies have reported rising PA from ages 3–6, with boys generally more active than girls across this span [3]. However, baseline findings from the Health Oriented Pedagogical Project (HOPP) suggest that PA declines during the primary school years [4]. Declining PA with increasing age has been observed across multiple settings and study designs [5,6,7,8]. Longitudinal and age-comparative evidence indicates that PA frequency and overall activity tend to decrease from late childhood through adolescence, with sex differences that may vary by age and context [9,10]. Broader syntheses also suggest that environmental and societal constraints contribute to reduced opportunities for children's activity [11]. Nordic studies similarly report lower activity at older ages and persistent sex differences, and some analyses imply that current recommendations may underestimate the amount of PA needed for optimal health [12,13]. More recent Norwegian data further indicate a concerning decline among specific subgroups, including 9-year-old boys, despite otherwise "fairly stable" levels over time [14].

Although some surveillance indicates modest secular improvements in children's mean PA in certain periods and contexts, the overall evidence base points to unfavorable age-related trajectories and rising sedentary behaviors that warrant continued monitoring [15,16]. Longitudinal evidence extending from adolescence into adulthood shows that very few individuals maintain favorable movement patterns over time, including simultaneously sustaining higher MVPA and lower screen time [17,18]. Cohort studies also document substantial declines in MVPA during adolescence, particularly among girls, alongside increases in leisure-time computer use, particularly among boys, as well as marked secular increases in screen-based behaviors [19]. Objective accelerometer data from early to late childhood show increasing sedentary

time and sedentary bouts, decreasing light-intensity physical activity (LPA), and moderate tracking of several movement metrics, suggesting that early-life patterns can persist and potentially compound across development [20].

These behavioral shifts have direct clinical relevance, as sedentary time and the full spectrum of movement behaviors are linked to trajectories in adiposity and cardiometabolic health from childhood into young adulthood [21]. Evidence from the ALSPAC cohort indicates that sedentary time is associated with adverse lipid profiles, while LPA may exert comparatively strong lipid benefits and may temporally precede later lipid changes, supporting the value of targeting the whole-day activity composition rather than MVPA alone [22]. Parallel findings suggest that reducing sedentary time and increasing LPA may be particularly important for attenuating insulin resistance risk across development, and that substituting sedentary time with LPA may yield meaningful cumulative blood-pressure benefits, with body composition, especially lean mass, playing an important confounding and mediating role [23,24]. This aligns with an emerging whole-day movement paradigm that emphasizes replacing sitting with standing and routine light activities, and with longitudinal evidence linking cumulative sedentary time to progressive adverse cardiac remodeling and LPA to more favourable cardiac structural changes during maturation [25,26].

The development of childhood activity change may be investigated with several design, as serial cross-sectional studies across years, as longitudinal trends, or as secular trends.. Several studies have used cross-sectional and secular trends, however fewer use longitudinal studies exist. The trend is clear regardless of design, as children grow older PA declines [27,7,28,29,30; 31,32].

Although the present research question is not novel, it addresses an essential public health need: societies should routinely monitor population-level physical activity (PA), including among children, to detect adverse trends early and implement timely countermeasures. Against this background, the aim of this study is to (i) describe the current level of PA in children and (ii) examine, using a longitudinal design, whether school-based PA interventions delivered as active learning embedded within the primary school curriculum can attenuate the age-related decline in children's PA over time.

## Methods

The Health Oriented Pedagogical Project (HOPP) was established in 2015 and implemented 45 min of additional, teacher-led physical activity integrated into the school day through active learning in all elementary schools in Horten municipality, Norway. The recruitment process and intervention are thoroughly described elsewhere [33].

### Study design

The Health Oriented Pedagogical Project was planned as a longitudinal (2015–2021) large-scale cohort study including a total population of n = 2816 participants with an age range of 6–12 years at baseline. The total study sample after recruitment and parental consent included n = 2297 children (82%). Seven elementary schools in Horten municipality, Norway, constitute the intervention group (n = 1545 at project start) with increased guided physical activity (active learning), and two control schools from the greater Oslo area (n = 752 at project start) having a normal activity level at school. Accelerometer data are collected each year in a set manner regarding the time of the year.

The original design encompassed data collection throughout the 2021 academic year. However, COVID-19 restricted school access in late spring 2020 and resulted in premature termination of data collection and cancellation of the 2021 measurement wave. In addition, a complete abandonment of the 2021 phase. The data was collected from 1–6 graders the first study year (2015) and 2–7 graders the following year (2016). Between 2017 and 2020, the number of grades, and thereby participants, were reduced due to transfer to secondary schools by one grade (approx. 350 participants) each year. The present analysis therefore uses data from 2015–2019 and includes participants with valid accelerometer data in 2015, comprising n = 1412 from intervention schools and n = 711 from control schools for a total baseline analytical sample of n = 2123. Test year was defined as the annual measurement wave (calendar year) and treated as time since baseline,

corresponding approximately to one year of ageing per wave and, in intervention schools, one additional year of intervention exposure.

## Physical activity assessment

Actigraph wGT3X-BT accelerometers (ActiGraph LLC, Pensacola, FL, USA) were used in measuring PA. Children were instructed to wear the accelerometer on the right hip for seven consecutive days during waking hours, removing it only when ill/absent or during water-based activities (e.g., showering or swimming). Sampling frequency was 100 Hz at 10 s epochs. A minimum of 8 h/day of registered activity was required for data analysis. Non-wear time was removed using the Troiano technique with 60 minutes of consecutive zeroes and 2 minutes of activity tolerance. For analyses, valid wear time was restricted to 06:00–23:59. Categorical division of PA levels was based on mean counts per minute (cpm) as sedentary (0–99 cpm), light (100–1999 cpm), moderate (2000–4999 cpm), and vigorous (≥5000 cpm), recording minutes in each intensity domain [34]. A combination of MVPA was calculated by summing minutes in moderate and vigorous intensity domains.

## Data analysis

Statistical analyses were performed in R (version 3.5.1) and NCSS 2024 v. 24.0.7 (NCSS LLC, Kaysville, UT, USA) using linear mixed-effects models to account for the hierarchical, repeated-measures structure of the data [35]. Fixed effects included test year, group (intervention vs control), and the group-by-time interaction to estimate differences in trajectories over time, with sex included as a covariate. Random intercepts were specified to account for non-independence due to repeated observations within children and clustering within schools and grades. The random-effects structure was selected a priori based on the study design and supported by model diagnostics and goodness-of-fit assessment, and we prioritized parsimonious specifications to ensure stable estimation. Random slopes for time were evaluated in sensitivity analyses, but these models did not materially improve model fit and showed signs of instability in some specifications, therefore results are reported from the random-intercept models. Statistical significance was set at $p < 0.05$. Cross-sectional analyses describe between-person differences across ages at a single time point, whereas longitudinal analyses estimate within-person change over time within the observed age span.

Children contributed all available valid accelerometer observations across measurement waves, and no imputation of missing outcome data was performed. Linear mixed-effects models were fitted using likelihood-based estimation with random intercepts for child ID to account for repeated measures and random intercepts for school and grade to account for clustering. This approach accommodates unbalanced repeated-measures data and provides valid inference under a Missing At Random assumption conditional on observed information included in the model. Observations with missing covariate data were excluded from the corresponding model fit (Fig 1).

## Socioeconomic status

The socioeconomic status (SES) in Norway is primarily defined by the combined parental education level. SES was defined from the lowest education level, primary/secondary school via high school, bachelor's degree and finally master's/PhD degree.

## Ethics and consent

The procedures and methods used in the study adhere to the ethical guidelines defined by the World Medical Association's Declaration of Helsinki. The Regional Committee for Medical Research Ethics (REK) has approved the research protocol (ref.no.: 2014/2064/REK). The research is catalogued in Clinical Trials (ClinicalTrials.gov Identifier: NCT02495714). Registered 20. June 2015. No personal data is published. Parents and legal guardians of all participants have provided informed consent.

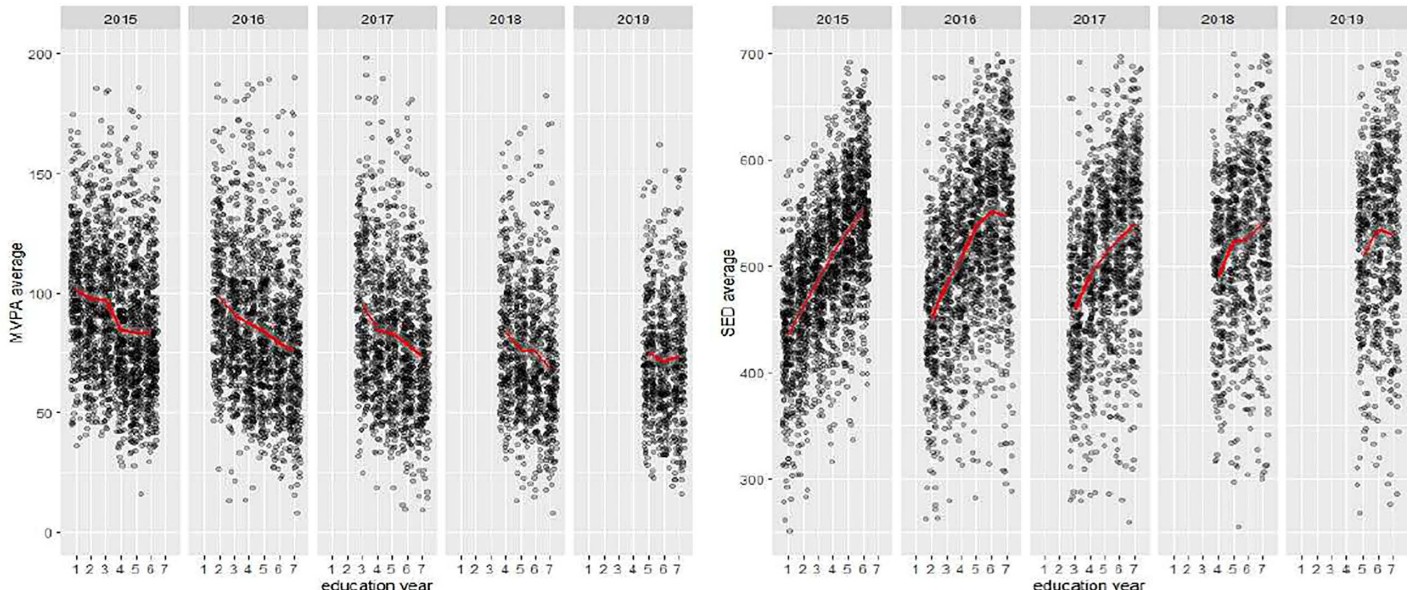

**Fig 1. Illustration of the distribution of moderate to vigorous physical activity (MVPA) and sedentary behaviour (SED) by test- and education years.**

## Results

### Longitudinal changes

A linear mixed model adjusted for sex showed that starting in 2015, with each additional intervention-year, time spent in MVPA declined on average by 2.2 min/day. Males were, on average, 12.4 min/day more active throughout the five years compared to females. The adjusted analysis also showed that the sedentariness of children increased by an average of 5.8 min/day over the years. Females, on average, had 14.2 min/day higher SED than males (WTable 1). To test whether trajectories differed between intervention and control schools, we fitted mixed-effects models including a group × test year term. Test year represents years since baseline (calendar time) and corresponds to annual within-child ageing during follow-up. The group × test year interaction was not statistically significant in the primary mixed model (Table 1), indicating limited evidence of different test-year–related slopes between groups. MVPA declined with test year in both groups; the estimated decline was steeper in controls than in the intervention group, but the difference in slopes was not statistically significant in this model (group × test year p = 0.24) (Table 1, Fig 2). We used the age of 10 as a reference, and at that age the intervention group had lower MVPA than the control group (adjusted difference −15.04 min/day; 95% CI −20.02 to −10.05; p < 0.001).

We used a sex-stratified piecewise approach with a group × test year specification, which suggested that the development of MVPA over time might differ by sex. For boys, there was no clear baseline difference in 2015 between intervention and control (−1.66 min/day; p = 0.309), but the trajectories diverged. The intervention group showed an attenuated post-2017 decline compared with controls (post-2017 slope ≈ −4.5 vs −11.4 min/day/year). For girls, there was a pronounced baseline imbalance in 2015 (intervention vs control −17.04 min/day; p < 0.001), while temporal trends were largely parallel; pre-2017 slopes were very similar and any post-2017 attenuation in the intervention group was smaller and more uncertain (female×intervention×T_post p = 0.091 in the current parameterization). Overall, the findings suggest that for boys the between-group differences are driven mainly by differences in change over time (especially after 2017), whereas for girls

**Table 1. Results of linear mixed model of moderate-to-vigorous physical activity (MVPA) and sedentary behavior (SED) for _longitudinal changes_.**

| Parameter | MVPA (min/day) | | | SED (min/day) | | |
|---|---|---|---|---|---|---|
| | Estimate (SE) | 95% CI | p | Estimate (SE) | 95% CI | p |
| Fixed intercept | 90.67 (2.30) | 86.16 \| 95.19 | < 0.001 | 438.31 (6.22) | 426.13 \| 450.50 | < 0.001 |
| Test year | −5.14 (0.54) | −6.20 \| −4.08 | < 0.001 | 17.74 (1.56) | 14.69 \| 20.79 | < 0.001 |
| Group (intervention) | 16.06 (3.37) | 9.45 \| 22.66 | < 0.001 | −4.80 (9.13) | −22.70 \| 13.09 | 0.60 |
| Group × test year | −1.10 (0.94) | −2.93 \| 0.74 | 0.24 | −2.36 (2.69) | −7.64 \| 2.92 | 0.38 |
| Sex (male) | 11.58 (2.11) | 7.44 \| 15.73 | < 0.001 | −9.30 (5.47) | −20.03 \| 1.42 | 0.09 |

CI – 95% confidence interval; SE – standard error. Test year is coded as years since 2015 (0 = 2015,..., 5 = 2020). Group indicates intervention vs control (control-intervention: 2 vs 1).

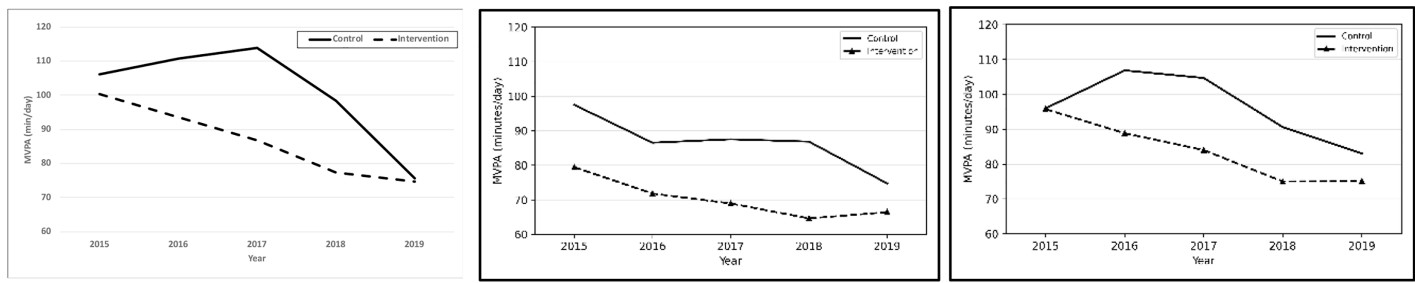

**Fig 2. The longitudinal results of moderate to vigorous physical activity (MVPA) for children born in 2008 who were six years old in 2015 are displayed.** The fall of MVPA in the control schools is more pronounced the three last years than in the intervention schools.

they are driven mainly by a lower starting level in the intervention group, with weaker evidence of differential slopes (Fig 3).

Fig 3 contrasts the age-related patterns estimated from the baseline cross-sectional data collected in 2015 with the within-person longitudinal trends observed across 2015–2019. For MVPA (left panel), the baseline cross-sectional age gradient indicated a steeper decline of approximately 3.5 min/day per year of age compared with the longitudinal estimate of 2.2 min/day per year during follow-up. For sedentary behavior (right panel), the baseline cross-sectional age gradient suggested a markedly steeper increase of approximately 22.5 min/day per year of age, whereas the longitudinal analyses showed an increase of approximately 5.7 min/day per year across 2015–2019. These lines are presented to illustrate differences in magnitude and direction between baseline cross-sectional gradients and longitudinal change estimates over time, rather than to imply that baseline slopes represent a causal counterfactual for subsequent within-person change.

## Discussion

### General findings

Males of all ages were more active than females by almost 12 min/day. The longitudinal analyses showed that time spent in MVPA by Norwegian 6–12-year-olds of both sexes declined from 2015 to 2019 by an average of 2.2 min·day⁻¹ per year, with males being more active throughout the 5-year period. This is less steep than the baseline cross-sectional age gradient presented for descriptive context but should not be interpreted as implying a one-to-one reallocation between MVPA and sedentary time. Furthermore, sedentary behavior increased by an average of 5.8 min·day⁻¹ per year. Changes in

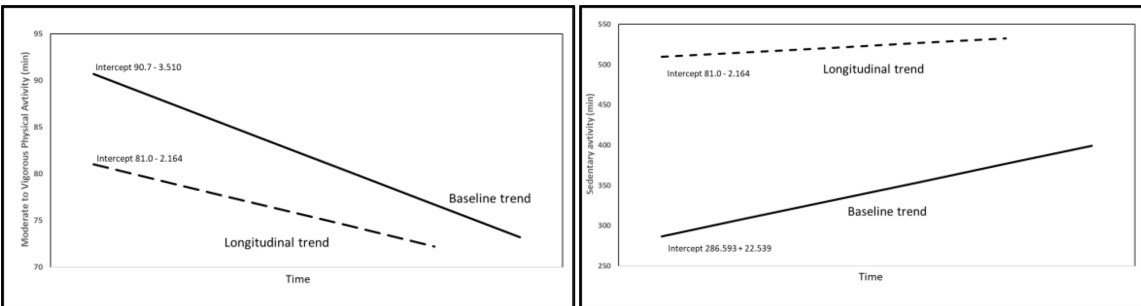

**Fig 3. Comparison of baseline and longitudinal trends across five years for moderate to vigorous physical activity (MVPA) and sedentary behaviour (SED).**

MVPA and sedentary time should be interpreted as distinct different behavioral constructs as time not spent in MVPA may be reallocated to light-intensity activity or standing rather than to sitting.

## Methodological considerations

In the present study, we examine how age-related differences derived from baseline cross-sectional data compare with longitudinal within-person changes observed in the same cohort across several age groups. Cross-sectional analyses spanning multiple age groups within a cohort are widely used because they are straightforward to implement, but they estimate an age gradient based on between-person differences at a single time point, whereas longitudinal analyses quantify within-person change over time and are generally considered more stable and informative for describing developmental trajectories. The heterogeneity in prior findings on whether PA increases or decreases with age may partly reflect limitations inherent to cross-sectional designs, including the inability to track individuals over extended periods. Importantly, we acknowledge that baseline cross-sectional slopes are not a causal counterfactual for longitudinal change; using cross-sectional age differences as a benchmark for "expected" longitudinal decline can be influenced by ecological and cohort effects and therefore cannot, on its own, support causal inference regarding intervention impact. Accordingly, in this manuscript we present baseline cross-sectional patterns as descriptive context, while the primary inference is based on longitudinal trajectories from 2015–2019, which provide the most robust evidence of change over time within the cohort.

The decrease in MVPA, on average, by 3.5 min·day$^{-1}$ per year was observed in 6–12-year-olds in a previous HOPP publication, using baseline cross-sectional data obtained in 2015, without a PA intervention [4]. The observed pattern was partly explained by the increase in mass and longer limbs causing a longer pendulum, both resulting in lower acceleration and, hence, a decline in measured PA by age (or rather growth). Despite this, there is a common understanding that children do reduce their PA with age, and today's school system increases this behaviour negatively through its pedagogical design. The HOPP study was designed to counteract the drop in PA with age by increasing PA during school classes through active learning.

Also, habitual, cultural, and environmental differences might add up to the observed discrepancies across several studies. The PA level is affected by several factors that some point to, for example, structural factors in society that prevent children from being as active as they want [11]. [9]. Also, organised PA was more important among the 13-year-olds, but they still were less active than 8-year-old children [9].

Reduction in PA with age in primary and secondary school may be affected by shifts toward more cognitively oriented activities, but this does not necessarily imply that reduced MVPA translates into equivalent increases in sedentary time. Some time it may shift toward light-intensity activity. Also, school teaching demands more sedentary behaviour of children, as does homework. Today's school teaching is mainly designed in a classroom setting, with limited room for PA. Results

from the HOPP study did indeed reveal that children have lower PA during school hours than during afternoons and weekends [36].

## Guidelines

Guidelines recommend a minimum of 60 min·day$^{-1}$ MVPA for children [37]. This does not need to occur in one continuous session but can be split into several intervals. Earlier published HOPP data has shown that up to 86% of the sample of elementary school children do achieve the recommended 60 min·day$^{-1}$ MVPA, averaging 90.7 min·day$^{-1}$, despite decreasing PA with age [4]. Even if the intervention in HOPP intended to increase the total amount of PA during school hours, a ceiling effect might have occurred in the intervention schools as the PA level was already high. This is illustrated in Fig 2, as all children born 2008, who participated in all five years, have MVPA above 60 min/day for both intervention and control schools, and any lack of effect from the intervention may be caused by a ceiling effect.

## MVPA

The present study reports longitudinal trends from 2015–2019 and relate these findings descriptively to the baseline cross-sectional patterns earlier reported from HOPP [4]. We acknowledge that cross-sectional baseline trends reflect between-person age differences at a single time point and therefore do not constitute a causal counterfactual for within-person longitudinal change. The baseline results revealed that time spent in MVPA decreased by 3.5 min/day for each year of age at baseline, showing a significant inverse correlation with age. After adjusting for factors such as age, BMI, waist circumference, parental education levels, and seasonal variations, they also found that boys were, on average, 8.9 min/day more active in MVPA per day than girls across all age groups.

The HOPP intervention did not aim to *increase* the PA for 6–12-years-olds, as it seemed highly unlikely that a limited increase in PA during school hours may counteract the increase in weight, height, and the social demand on children to move less. Instead, the intervention had the goal of reducing the annual decline. The main goal of HOPP was to reduce the decline to less than 3.5 min·day$^{-1}$ per year and interpret this as a positive effect of the intervention. Any physical activity (PA) intervention that successfully mitigates this decline would be considered effective. Additionally, initiatives that enhance PA in children's daily routines without necessarily reversing a decline in activity levels would also be seen as beneficial. This is important because children's overall activity levels are influenced by more than just school-based programs; activities before and after school, as well as on weekends, significantly affect overall PA. Data from HOPP has earlier showed that the drop in activity is mainly caused by a decline in school hours [38]. Therefore, increasing PA in school hours, particularly through active learning may reduce the age-related decline in PA.

Juxtaposing the longitudinal within-person estimate from 2015–2019 with the baseline cross-sectional age gradient suggests a smaller decline in MVPA during follow-up, with the longitudinal slope of 2.2 min·day$^{-1}$·year$^{-1}$ compared with 3.5 min·day$^{-1}$·year$^{-1}$ at baseline, as shown in Fig 2. Importantly, these quantities are not directly comparable in a causal sense, because cross-sectional age differences reflect between-person variation at one time point and do not provide a causal counterfactual for within-person longitudinal change, and they may be influenced by cohort and ecological effects. With this limitation in mind, the contrast corresponds to a difference of 1.3 min·day$^{-1}$·year$^{-1}$ and may be used to contextualize the potential practical significance over time. An average annual reduction of 3.5 min·day$^{-1}$ from age 6 would accumulate, if linearity and stability is maintained, to 17 min·day$^{-1}$ less MVPA by age 12, corresponding to 103 hours less MVPA across one year at age 12. By comparison, a decline of 2.2 min·day$^{-1}$ would accumulate to 11 min·day$^{-1}$ and 66 hours less MVPA across one year at age 12, equaling a difference of 37 hours per year. These calculations are provided to illustrate magnitude and public health relevance rather than to establish causality, and interpretation of intervention impact is therefore based on the longitudinal change estimates presented in this study.

We tested whether intervention and control schools differed in longitudinal trajectories by including a group × time interaction in the mixed-effects models. This indicated that MVPA declined with age in both groups, but the decline was less steep in the intervention group than in controls. At the same time, the intervention group had lower MVPA at the reference age, consistent with baseline between-group differences and the potential influence of contextual factors such as socioeconomic composition. The sex-stratified analyses indicate that changes over time were not homogeneous: the sharper post-2017 decline observed in the control group appears to be driven mainly by boys, while girls showed comparatively more stable and parallel trends. This highlights the importance of considering sex-specific trajectories when interpreting group differences over time and suggests that contextual factors may differentially affect boys' and girls' activity patterns during late primary school.

While the trajectory difference is consistent with attenuation of age-related decline in the intervention schools, the non-randomized design and baseline differences mean the findings should be interpreted as an association rather than definitive proof of a causal intervention effect.

### Sedentary behaviour

For sedentary behavior, the baseline cross-sectional age gradient suggested an increase of 22.5 min·day$^{-1}$·year$^{-1}$ between ages 6 and 12 years. [4]. In the present longitudinal analyses, sedentary time increased by 5.7 min·day$^{-1}$·year$^{-1}$. These estimates are shown for context only, as baseline cross-sectional age differences are not directly comparable with within-person longitudinal change and may reflect cohort effects. With this limitation in mind, the shallower longitudinal slope is consistent with a potentially reduced tendency toward increasing sedentary time with age during the follow-up period. Regardless of the comparison, the observed increase in sedentary behavior is concerning. These findings support continued follow-up of children and adolescents to mitigate the development of inactivity with age and suggest that school-based intervention programs may be one feasible strategy with the potential to contribute positively.

### Socioeconomic status

The initial analyses from baseline revealed that the control schools (96.7 min·day$^{-1}$) had a higher MVPA than the intervention schools (87.7 min·day$^{-1}$) to start with (p < 0.0001). The differences have not been largely reduced, as our data still show higher MVPA in control schools. This discrepancy may partly be explained by higher parental socioeconomic status (SES) in the control schools compared to intervention schools. In Norway, where income and access to material goods are relatively equally distributed across the population, high SES is predominantly associated with higher education. These differences might influence the results of PA among children, as children from rural parts were less active than children from urban areas [9]. Individuals with higher education frequently relocate closer to major urban centres, and the two control schools involved are situated near Oslo, the capital of Norway.

### Comparison with previous studies

The present HOPP analyses showed a clear age-related deterioration with MVPA declining and sedentary behavior increasing across 6–12 year olds. This alongside sex difference where boys accumulated more MVPA, and girls accumulated more sedentary time. These findings align with earlier studies and are typical developmental patterns rather than anomalies restricted to single settings [7; 20]. The decline in in PA appears to be driven largely by decreasing light-intensity activity and increasing sedentary time [27; 5; 32]. This underscores that changes in MVPA and sedentary time are not simple inverses because time may be reallocated between sitting and light-intensity activities without necessarily producing proportional changes in MVPA.

Evidence suggests that the decline starts well before adolescence, as shown in the present study as in others, PA trajectories declined from age 6 years, implying the adolescence is not the primary onset of decline [8]. Earlier Norwegian

data also demonstrate substantial adverse longitudinal change from age 9–15 years, including reductions in MVPA and large increases in sedentary time [7]. Similar conclusions emerge from self-report cohorts spanning adolescence into young adulthood, where most participants fail to maintain favorable MVPA levels over time and screen-based behaviors increase [18; 19].

A consistent implication of this literature is that early patterns tend to persist and therefore represent a plausible leverage point for prevention. Objective studies show moderate tracking of PA and sedentary time from childhood to adolescence, particularly when major sources of variation are accounted for, suggesting that relative activity ranking is not entirely transient [13]. Longitudinal accelerometer studies further indicate that sedentary time and sedentary bout accumulation increase from early to late childhood while light-intensity PA decreases, and that many movements metrics track across multiple measurement waves [20]. From a health perspective, these behavioral shifts are clinically salient because sedentary time and the composition of daily movement behaviors are associated with long-term trajectories in adiposity and cardiometabolic risk; longitudinal evidence links higher sedentary time to increased fat mass and adverse lipid profiles, whereas higher light activity and MVPA are generally associated with more favorable risk profiles [23; 21]. Related work also suggests that replacing sedentary time with light activity may yield meaningful cumulative benefits for blood pressure and may be an important target for insulin resistance prevention, emphasizing the broader relevance of reducing sedentariness across development [22, 24].

Against this backdrop, the HOPP findings contribute to the growing argument that counteracting age-related declines requires structural solutions that operate during the school day, rather than relying solely on leisure-time sport participation. Constraints linked to school policy and curricula have been highlighted as plausible contributors to declining activity opportunities, supporting the rationale for embedding movement within ordinary teaching [11]. Active learning represents one such structural approach which conceptually aligned with the "whole day matters" perspective that promotes more activity replacing sitting [26]. The HOPP intervention trajectories, where the decline in MVPA in intervention schools appeared to level off compared to control schools, are consistent with the proposition that curriculum-embedded activity can attenuate, the decline in MVPA. Continued surveillance remain warranted as many countries still report that a many children do not meet PA recommendations and exhibit high sedentary behavior, reinforcing the need for scalable school-based strategies that can be sustained over time [39].

## Strength and limitations

The principal strengths of this study include its large sample size, the objective measurement of physical activity (PA) using accelerometers, and its longitudinal design, which tracks the same children over a period of five years, enabling the analysis of long-term trends.

## Conclusion

The longitudinal analysis in the HOPP study indicates that active learning may help attenuate the age-related decline in children's physical activity. Baseline cross-sectional age differences suggested a decline of 3.5 min/day per year of age in MVPA. During follow-up (2015–2019), MVPA decreased by an average of 2.2 min/day per year, and this longitudinal estimate was smaller than the baseline cross-sectional age gradient presented for context rather than as a causal counterfactual. Sedentary behavior increased over time, although the observed longitudinal slope was less steep than the baseline cross-sectional age gradient used for descriptive context, and the increase remains concerning.

## Trial registration

Clinical Trial.gov. Identifier: NCT02495714. Registered 20 June 2015. https://register.clinicaltrials.gov/prs/app/action/LogoutUser?uid=U0002ORK&ts=13&sid=S0005MCN&cx=pc85td.

## Author contributions

**Conceptualization:** Per Morten Fredriksen, Asgeir Mamen.

**Data curation:** Per Morten Fredriksen, Maren Valand Hæhre, Asgeir Mamen.

**Formal analysis:** Per Morten Fredriksen, Iana Kharlova, Asgeir Mamen.

**Funding acquisition:** Per Morten Fredriksen, Asgeir Mamen.

**Investigation:** Per Morten Fredriksen.

**Methodology:** Per Morten Fredriksen, Maren Valand Hæhre.

**Project administration:** Per Morten Fredriksen, Maren Valand Hæhre.

**Writing – original draft:** Per Morten Fredriksen, Iana Kharlova, Maren Valand Hæhre, Asgeir Mamen.

**Writing – review & editing:** Per Morten Fredriksen, Iana Kharlova, Maren Valand Hæhre, Asgeir Mamen.

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
