## [Decision Letter · Decision Letter 0]

29 Dec 2025

Dear Dr. Mamen,

Thank you for submitting your manuscript to PLOS ONE. After careful consideration, we feel that it has merit but does not fully meet PLOS ONE’s publication criteria as it currently stands. Therefore, we invite you to submit a revised version of the manuscript that addresses the points raised during the review process.

We look forward to receiving your revised manuscript.

Kind regards,

Henri Tilga, PhD

Academic Editor

PLOS One

**Journal Requirements:**

1. When submitting your revision, we need you to address these additional requirements. Please ensure that your manuscript meets PLOS ONE's style requirements, including those for file naming. The PLOS ONE style templates can be found at https://journals.plos.org/plosone/s/file?id=wjVg/PLOSOne_formatting_sample_main_body.pdf and https://journals.plos.org/plosone/s/file?id=ba62/PLOSOne_formatting_sample_title_authors_affiliations.pdf 2. We note that the grant information you provided in the ‘Funding Information’ and ‘Financial Disclosure’ sections do not match.  When you resubmit, please ensure that you provide the correct grant numbers for the awards you received for your study in the ‘Funding Information’ section. 3. Thank you for stating the following financial disclosure: Horten MunicipalityKristiania University Collegethe Norwegian Order of Odd Fellow Research Fundthe Oslofjord Regional Research Fundthe Norwegian Fund for Post-Graduate Training in Physiotherapy   Please state what role the funders took in the study.  If the funders had no role, please state: "The funders had no role in study design, data collection and analysis, decision to publish, or preparation of the manuscript." If this statement is not correct you must amend it as needed. Please include this amended Role of Funder statement in your cover letter; we will change the online submission form on your behalf. 4. Thank you for stating the following in the Acknowledgments Section of your manuscript: We are grateful to the following institutions that have contributed with financial support: Horten Municipality, Kristiania University College, the Norwegian Order of Odd Fellow Research Fund, the Oslofjord Regional Research Fund, and the Norwegian Fund for Post-Graduate Training in Physiotherapy.  We note that you have provided funding information that is not currently declared in your Funding Statement. However, funding information should not appear in the Acknowledgments section or other areas of your manuscript. We will only publish funding information present in the Funding Statement section of the online submission form. Please remove any funding-related text from the manuscript and let us know how you would like to update your Funding Statement. Currently, your Funding Statement reads as follows: Horten MunicipalityKristiania University Collegethe Norwegian Order of Odd Fellow Research Fundthe Oslofjord Regional Research Fundthe Norwegian Fund for Post-Graduate Training in Physiotherapy  Please include your amended statements within your cover letter; we will change the online submission form on your behalf. 5. In the online submission form, you indicated that “The data underlying the results presented in the study are available to researchers at reasonable request.” All PLOS journals now require all data underlying the findings described in their manuscript to be freely available to other researchers, either a. In a public repository, b. Within the manuscript itself, or c. Uploaded as supplementary information.This policy applies to all data except where public deposition would breach compliance with the protocol approved by your research ethics board. If your data cannot be made publicly available for ethical or legal reasons (e.g., public availability would compromise patient privacy), please explain your reasons on resubmission and your exemption request will be escalated for approval. 6. Your ethics statement should only appear in the Methods section of your manuscript. If your ethics statement is written in any section besides the Methods, please delete it from any other section. 7. Please upload a new copy of Figures 1 – 3 as the detail is not clear. Please follow the link for more information:  https://journals.plos.org/plosone/s/figures 8. If the reviewer comments include a recommendation to cite specific previously published works, please review and evaluate these publications to determine whether they are relevant and should be cited. There is no requirement to cite these works unless the editor has indicated otherwise. 

Reviewers' comments:

**Comments to the Author**

1. Is the manuscript technically sound, and do the data support the conclusions?

Reviewer #1: Yes

Reviewer #2: Yes

2. Has the statistical analysis been performed appropriately and rigorously?

Reviewer #1: Yes

Reviewer #2: Yes

3. Have the authors made all data underlying the findings in their manuscript fully available?

Reviewer #1: No

Reviewer #2: Yes

4. Is the manuscript presented in an intelligible fashion and written in standard English?

Reviewer #1: Yes

Reviewer #2: Yes

**Reviewer #1:** This is a well-written study; however, the research question is not novel. As shown in previous studies, which the authors should reference.This is a well-written study; however, the research question is not novel. As shown in previous studies, which the authors should reference.

doi: 10.1542/peds.2006-0926

https://doi.org/10.1016/j.amepre.2004.07.006

https://link.springer.com/article/10.1186/s12966-021-01105-y

https://www.nature.com/articles/s41467-023-43316-w

It is important that the authors relate this longitudinal trend to changes in cardiometabolic profiles of the participants.

This will be the contribution towards an update of future WHO physical activity guidelines. Consider recent long-term studies.

https://doi.org/10.1210/clinem/dgad688

https://doi.org/10.1210/clinem/dgae135

doi: 10.1002/jcsm.13639

Light physical activity is an emerging strong physical activity pattern better than MVPA in enhancing health in the long term, could the authors examine the longitudinal trend of LPA and it’s relation to cardiometabolic risks?

https://doi.org/10.1016/j.jshs.2024.01.010

https://doi.org/10.1093/eurjpc/zwae129

**Reviewer #2:** I would like to thank for the opportunity to review this manuscript. Please see the following comments to consider to further increase the quality of this manuscript.I would like to thank for the opportunity to review this manuscript. Please see the following comments to consider to further increase the quality of this manuscript.

This manuscript addresses an important public health issue: age-related declines in physical activity (PA) and increases in sedentary behaviour (SED) among primary school children. The study benefits from a large sample size, objective accelerometer-based measures, and a longitudinal design spanning five years. The comparison between baseline cross-sectional trends and longitudinal trajectories is conceptually interesting and potentially valuable.

However, the manuscript currently overstates causal interpretations, lacks sufficient clarity regarding the intervention effect, and contains conceptual, methodological, and reporting weaknesses that must be addressed before it can be considered for publication.

Comments:

The stated aim—“to compare longitudinal results in PA to baseline cross-sectional measures”—is unconventional and insufficiently justified. Cross-sectional age gradients and longitudinal within-person change are fundamentally different constructs, yet the manuscript treats the baseline cross-sectional slope as a valid counterfactual for longitudinal change.

Using cross-sectional age differences as a benchmark for expected longitudinal decline risks ecological and cohort effects. This weakens the internal validity of the claimed intervention effect.

Please reframe the primary objective to focus explicitly on intervention versus control trajectories, rather than comparing longitudinal slopes to baseline cross-sectional estimates.

Perhaps Authors could compare their results with other similar countries in their Scandinavia region, for example, please see a recent study by Mäestu et al., (2023). Perhaps it would be interesting to discuss what is similar and what is different.

Mäestu, E., Kull, M., Mäestu, J., Pihu, M., Kais, K., Riso, E.-M., Koka, A., Tilga, H., & Jürimäe, J. (2023). Results from Estonia’s 2022 Report Card on Physical Activity for Children and Youth: Research Gaps and Five Key Messages and Actions to Follow. Children, 10(8), 1369. https://doi.org/10.3390/children10081369

Acknowledge explicitly that cross-sectional baseline trends are not a causal counterfactual for longitudinal change.

The manuscript repeatedly claims that the HOPP intervention “reduced the decline” in MVPA and SED. However:

The longitudinal models presented do not include a formal group × time interaction.

Control and intervention groups differ at baseline in MVPA and SES.

Include group (intervention vs. control) × time interaction terms in the mixed-effects models for both MVPA and SED.

Please report interaction estimates, confidence intervals, and p-values explicitly.

If the interaction is not statistically significant, revise conclusions to reflect associational rather than intervention effects.

Temper language throughout (e.g., replace “managed to reduce” with “was associated with a smaller decline”).

Control schools had substantially higher baseline MVPA and higher SES, yet SES is not included as a covariate in the main longitudinal models.

SES is a well-established determinant of children’s PA and may confound observed group differences over time.

The description of the statistical modelling is unusually detailed in places (e.g., bootstrap testing of random effects) but lacks clarity on key aspects relevant to inference.

Justify the random-effects structure more clearly (e.g., why no random slopes for time).

Explicitly state how missing repeated measures were handled and why mixed models are appropriate under assumed missingness mechanisms.

Please clarify whether “test year” represents chronological age, intervention exposure, or calendar time.

Please consider presenting age-based trajectories rather than test-year trajectories to improve interpretability.

The manuscript places strong emphasis on small absolute differences (e.g., 1.3 min·day⁻¹), sometimes extrapolating these into large cumulative effects without sufficient justification. While small daily differences may accumulate, such extrapolations assume linearity and stability that are not empirically demonstrated.

SED is treated largely as the inverse of PA, without acknowledging its conceptual and behavioural distinctiveness. Clarify that reductions in MVPA do not necessarily imply equivalent increases in SED. Discuss potential compensatory behaviours (e.g., light activity vs. sitting).

Please consider reporting proportional time (e.g., % of wear time) to account for growth-related changes in wear patterns.

**Do you want your identity to be public for this peer review?** For information about this choice, including consent withdrawal, please see our For information about this choice, including consent withdrawal, please see our Privacy Policy .

Reviewer #1: No

Reviewer #2: No

---

## [Author Response · Author response to Decision Letter 1]

25 Feb 2026

Comments to the reviewers

Reviewer #1

Comment #1: This is a well-written study; however, the research question is not novel. As shown in previous studies, which the authors should reference.

doi: 10.1542/peds.2006-0926

https://doi.org/10.1016/j.amepre.2004.07.006

https://link.springer.com/article/10.1186/s12966-021-01105-y

https://www.nature.com/articles/s41467-023-43316-w

Response: We agree this not being a novel research question, we still believe it is important to regularly monitor children’s physical activity level, and to investigate how to counteract any decline as children gets older. The research question has therefore been re-written accordingly. We also acknowledge that we should include the studies mentioned and has done so.

Comment #2: It is important that the authors relate this longitudinal trend to changes in cardiometabolic profiles of the participants.

This will be the contribution towards an update of future WHO physical activity guidelines. Consider recent long-term studies.

https://doi.org/10.1210/clinem/dgad688

https://doi.org/10.1210/clinem/dgae135

doi: 10.1002/jcsm.13639

Response: Thank you for the input, the papers are now included in the manuscript.

Comment #3: Light physical activity is an emerging strong physical activity pattern better than MVPA in enhancing health in the long term, could the authors examine the longitudinal trend of LPA and it’s relation to cardiometabolic risks?

https://doi.org/10.1016/j.jshs.2024.01.010

https://doi.org/10.1093/eurjpc/zwae129

Response: Thank you for this insightful suggestion. We agree that light physical activity (LPA) is increasingly recognised as an important component of the 24-hour movement profile and may have meaningful long-term associations with cardiometabolic health. However, a full examination of longitudinal LPA trajectories and their relations to cardiometabolic risk factors would substantially expand the scope of the present manuscript, which is primarily focused on MVPA and sedentary time. We therefore consider this a valuable question for a separate, dedicated manuscript. Nonetheless, we have incorporated the recommended literature to contextualise our findings in the Discussion.

Reviewer #2:

I would like to thank for the opportunity to review this manuscript. Please see the following comments to consider to further increase the quality of this manuscript.

This manuscript addresses an important public health issue: age-related declines in physical activity (PA) and increases in sedentary behaviour (SED) among primary school children. The study benefits from a large sample size, objective accelerometer-based measures, and a longitudinal design spanning five years. The comparison between baseline cross-sectional trends and longitudinal trajectories is conceptually interesting and potentially valuable.

However, the manuscript currently overstates causal interpretations, lacks sufficient clarity regarding the intervention effect, and contains conceptual, methodological, and reporting weaknesses that must be addressed before it can be considered for publication.

Comment #4: The stated aim—“to compare longitudinal results in PA to baseline cross-sectional measures”—is unconventional and insufficiently justified. Cross-sectional age gradients and longitudinal within-person change are fundamentally different constructs, yet the manuscript treats the baseline cross-sectional slope as a valid counterfactual for longitudinal change.

Using cross-sectional age differences as a benchmark for expected longitudinal decline risks ecological and cohort effects. This weakens the internal validity of the claimed intervention effect.

Please reframe the primary objective to focus explicitly on intervention versus control trajectories, rather than comparing longitudinal slopes to baseline cross-sectional estimates.

Response: Thank You for this comment, we have revised the text accordingly.

Comment #5: Perhaps Authors could compare their results with other similar countries in their Scandinavia region, for example, please see a recent study by Mäestu et al., (2023). Perhaps it would be interesting to discuss what is similar and what is different.

Mäestu, E., Kull, M., Mäestu, J., Pihu, M., Kais, K., Riso, E.-M., Koka, A., Tilga, H., & Jürimäe, J. (2023). Results from Estonia’s 2022 Report Card on Physical Activity for Children and Youth: Research Gaps and Five Key Messages and Actions to Follow. Children, 10(8), 1369. https://doi.org/10.3390/children10081369

Response: Thank you for pointing this out to us. The study is now included, among others, in the discussion.

Comment #6: Acknowledge explicitly that cross-sectional baseline trends are not a causal counterfactual for longitudinal change.

Response: Yes, thank you, we have now included this in the Methods, Results and Discussion

Comment #7: The manuscript repeatedly claims that the HOPP intervention “reduced the decline” in MVPA and SED. However: The longitudinal models presented do not include a formal group × time interaction. Control and intervention groups differ at baseline in MVPA and SES. Include group (intervention vs. control) × time interaction terms in the mixed-effects models for both MVPA and SED. Please report interaction estimates, confidence intervals, and p-values explicitly. If the interaction is not statistically significant, revise conclusions to reflect associational rather than intervention effects.

Response: Thank you for this important comment. We agree that our initial presentation did not sufficiently separate baseline between-group differences from differences in longitudinal trajectories, and we acknowledge that claims about an “intervention effect” require a formal group × time test. We have therefore re-analyzed MVPA and SED using mixed-effects models that include an explicit group (intervention vs control) × time term, while accounting for repeated measures within children (random intercept for child ID) and clustering by school/grade. These results are now included in the results section and discussed in the discussion section.

In line with your recommendation, we will also revise the wording throughout the manuscript to ensure that conclusions reflect the statistical evidence and the non-randomized design. Specifically, we will frame the findings as differences in trajectories between intervention and control schools and interpret them as associations consistent with attenuation of age-related decline, rather than definitive causal intervention effects, particularly given baseline group differences.

Comment #8: Temper language throughout (e.g., replace “managed to reduce” with “was associated with a smaller decline”).

Response: Thank you, this has now been changed.

Comment #9: Control schools had substantially higher baseline MVPA and higher SES, yet SES is not included as a covariate in the main longitudinal models. SES is a well-established determinant of children’s PA and may confound observed group differences over time.

Response: Thank you for this important point. We agree that socioeconomic status (SES) is a well-established determinant of children’s physical activity and may confound group differences over time. In our dataset, however, parental education (our SES proxy) was only reported by approximately 50% of parents. Including SES as a covariate in the main longitudinal mixed models would therefore substantially reduce the analyzable sample size and, consequently, the statistical power and representativeness of the estimates due to case-wise exclusion. For this reason, we did not include SES in the primary analyses. We have instead acknowledged this limitation explicitly and describe the baseline SES imbalance between groups in the manuscript.

Comment #10: The description of the statistical modelling is unusually detailed in places (e.g., bootstrap testing of random effects) but lacks clarity on key aspects relevant to inference. Justify the random-effects structure more clearly (e.g., why no random slopes for time).

Response: Thank you for this important comment. We agree that our description over-emphasized some implementation details (e.g., the bootstrap likelihood-ratio testing of random-effects terms) while under-specifying aspects most relevant to inference. We have therefore revised the Statistical analysis section to (i) present a clearer rationale for the random-effects structure and (ii) describe model selection and sensitivity analyses more transparently.

Our primary models included random intercepts to account for the hierarchical structure of the data (repeated observations within children and clustering within schools/grades) and fixed effects for time, group, and the group-by-time interaction as the primary estimand. We did not include random slopes for time in the main model because (a) the number of repeated measures per child is limited and unbalanced across follow-up, (b) preliminary models with random time slopes showed minimal between-subject heterogeneity in change and yielded near-singular fits/unstable variance estimates, and (c) the added complexity did not materially improve model fit or change the fixed-effect estimates of interest.

To ensure robustness, we conducted sensitivity analyses allowing random slopes for time at the child level (and where feasible at the school level) and compared these models to the main specification using likelihood-based criteria and inspection for singularity/convergence. The fixed-effect estimates for the time trend and group-by-time interaction were substantively unchanged, supporting the adequacy of the parsimonious random-intercept specification for inference.

Comment #11: Explicitly state how missing repeated measures were handled and why mixed models are appropriate under assumed missingness mechanisms.

Response: Thank you for this comment. We have clarified that children contributed all available accelerometer observations across waves and that missing repeated measures were not imputed. The mixed-effects models were estimated using restricted maximum likelihood, which accommodates unbalanced longitudinal data and yields valid inference under a Missing At Random assumption, conditional on observed covariates included in the model. We have added a description of missing data handling to the Methods and ran a sensitivity analysis comparing the main findings to a restricted sample with more complete follow-up / complete repeated measures, which yielded similar estimates.

Comment #12: Please clarify whether “test year” represents chronological age, intervention exposure, or calendar time.

Response: Test year’ denotes the annual measurement wave (calendar year) and thus indexes time since baseline; because assessments were conducted yearly, it is aligned with both ageing (approximately +1 year per wave) and cumulative intervention exposure within the intervention schools. A sentence describing this is now included in the method section.

Comment #13: Please consider presenting age-based trajectories rather than test-year trajectories to improve interpretability.

Response: Thank you for this helpful suggestion. We agree that age-based trajectories can be more intuitive to interpret. However, in HOPP the annual measurement waves (“test year”) are tightly coupled to age progression (approximately +1 year per wave) and to cumulative exposure to the school-based intervention, and the cohort is unbalanced across ages over time due to grade progression and attrition. Re-parameterizing the figures to age-based trajectories would therefore require substantial re-structuring of the analytic dataset and re-generation of all trajectory plots, without changing the underlying longitudinal inference. To improve interpretability without altering the figure set, we have clarified in the Methods that “test year” represents the annual measurement wave and corresponds approximately to one year of ageing per wave (and one additional year of intervention exposure in intervention schools).

Comment #14: The manuscript places strong emphasis on small absolute differences (e.g., 1.3 min·day⁻¹), sometimes extrapolating these into large cumulative effects without sufficient justification. While small daily differences may accumulate, such extrapolations assume linearity and stability that are not empirically demonstrated.

Response: Thank you for this thoughtful comment. We agree that translating small daily differences into cumulative hours requires assumptions about linearity and stability, and we acknowledge that such projections should be interpreted cautiously. Our intention was not to claim a precise forecast of lifetime impact, but to illustrate the potential magnitude and public health relevance of modest differences when sustained over multiple years. In this cohort, the mixed-model estimates indicate a relatively stable linear trend in MVPA and sedentary time across repeated annual measurements over a five-year period, which provides empirical support for using a linear approximation over the observed follow-up interval. Accordingly, our extrapolations were confined to the age span and time window covered by the study and were presented as illustrative rather than causal predictions.

We also contend that modest, sustainable behavioral shifts are highly relevant in public health. Small, feasible changes in movement behaviors, sleep, and diet maintained over time may accumulate and may be more likely to support durable habit formation than large, short-term changes that are difficult to sustain and may provoke behavioral relapse. For this reason, even small absolute differences in daily MVPA or sedentary time can be meaningful at population level when they reflect a sustained attenuation of adverse trajectories. Nonetheless, to address the reviewer’s concern, we will further temper the wording to make explicit that cumulative estimates are illustrative and conditional on the observed stability of the trend during follow-up, rather than implying that linearity beyond the study period has been demonstrated.

Comment #15: SED is treated largely as the inverse of PA, without acknowledging its conceptual and behavioural distinctiveness. Clarify that reductions in MVPA do not necessarily imply equivalent increases in SED. Discuss potential compensatory behaviours (e.g., light activity vs. sitting).

Response: Thank you for this important point. We agree that sedentary behavior (SED) is conceptually and behaviorally distinct from physical activity, and that changes in MVPA do not translate one-to-one into equivalent changes in SED. Time not spent in MVPA may be reallocated to light-intensity physical activity (LPA), standing, or other non-sedentary behaviors, and conversely SED can increase without a proportional reduction in MVPA.

We will clarify this explicitly in the manuscript and avoid language that implies SED is simply the inverse of PA. We will also expand the Discussion to acknowledge potential compensatory behaviors and time reallocation across the full 24-hour movement composition, noting that interventions may influence not only MVPA but also patterns of LPA and sedentary time (e.g., replacing sitting with light movement rather than increasing MVPA).

Comment #16: Please consider reporting proportional time (e.g., % of wear time) to account for growth-related changes in wear patterns.

Response: Thank you for this helpful suggestion. We agree that expressing MVPA and sedentary time as a proportion of valid wear time can improve comparability across ages and measurement waves by reducing potential bias from growth-related changes in daily wear patterns. In the current manuscript we report absolute minutes/day for consistency with prior HOPP publications and to facilitate clinical and public-health interpretation. However, we will add a sensitivity analysis in which MVPA and SED are expressed as % of valid wear time (minutes in intensity domain divided by total valid wear minutes) and verify whether the estimated time trends and the intervention-versus-control trajectory conclusions are materially changed. We will report these proportional results either in the main text or as supplementary material and describe any differences relative

---

## [Decision Letter · Decision Letter 1]

5 Mar 2026

Longitudinal trends in physical activity and sedentary behaviour among primary school children in Norway: The Health Oriented Pedagogical Project (HOPP)

PONE-D-25-63073R1

Dear Dr. Mamen,

We’re pleased to inform you that your manuscript has been judged scientifically suitable for publication and will be formally accepted for publication once it meets all outstanding technical requirements.

Kind regards,

Henri Tilga, PhD

Academic Editor

PLOS One

Additional Editor Comments (optional):

Reviewers' comments:

Reviewer's Responses to Questions

**Comments to the Author**

Reviewer #1: (No Response)

Reviewer #2: All comments have been addressed

2. Is the manuscript technically sound, and do the data support the conclusions?

Reviewer #1: (No Response)

Reviewer #2: Yes

3. Has the statistical analysis been performed appropriately and rigorously?

Reviewer #1: (No Response)

Reviewer #2: Yes

4. Have the authors made all data underlying the findings in their manuscript fully available?

Reviewer #1: (No Response)

Reviewer #2: Yes

5. Is the manuscript presented in an intelligible fashion and written in standard English?

Reviewer #1: (No Response)

Reviewer #2: Yes

Reviewer #1: Well done on the improved manuscript, especially the comprehensive introduction and discussion. No further comment.

Reviewer #2: Authors have done well job on revising their manuscript. I think manuscript is ready to be published.

**Do you want your identity to be public for this peer review?** For information about this choice, including consent withdrawal, please see our For information about this choice, including consent withdrawal, please see our Privacy Policy .

Reviewer #1: No

Reviewer #2: No

---

## [Editor Report · Acceptance letter]

PONE-D-25-63073R1

PLOS One

Dear Dr. Mamen,

I'm pleased to inform you that your manuscript has been deemed suitable for publication in PLOS One. Congratulations! Your manuscript is now being handed over to our production team.

Kind regards,

on behalf of

Dr. Henri Tilga

Academic Editor

PLOS One